# Farmers' willingness to pay for foot and mouth disease vaccine in different cattle production systems in Amhara region of Ethiopia

**Wudu T. Jemberu**[1]*, **Wassie Molla**[1], **Tigabu Dagnew**[2], **Jonathan Rushton**[3], **Henk Hogeveen**[4]

**1** Department of Veterinary Epidemiology and Public Health, College of Veterinary Medicine and Animal Sciences, University of Gondar, Gondar, Ethiopia, **2** Department of Agricultural Economics, College of Agriculture and Environmental Sciences, University of Gondar, Gondar, Ethiopia, **3** Institute of Infection and Global Health, University of Liverpool, Liverpool, United Kingdom, **4** Business Economics Group, Wageningen University and Research, Wageningen, The Netherlands

* Wudu.temesgen@uog.edu.et, wudutemesgen@gmail.com

**Data Availability Statement:** All relevant data are within the manuscript and its Supporting Information files.

## Abstract

Although foot and mouth disease (FMD) is endemic in Ethiopia, use of vaccines to control the disease has been practiced sparingly. This is due to perceived high cost of good quality FMD vaccine, and consequently limited availability of the vaccine in the market. This study was conducted to assess farmers' willingness to pay (WTP) for a quality FMD vaccine and identify factors that could potentially influence their WTP in Amhara region of Ethiopia. A total of 398 farmers from four districts that represent the mixed crop-livestock and market oriented production systems were enrolled for the study. The WTP was estimated using contingent valuation method with a double-bound dichotomous choice bid design. Interval regression analysis was used to estimate mean WTP and identify factors that influence it. The results showed that the mean WTP of all farmers was Ethiopian Birr (ETB) 58.23 (95% CI: 56.20–60.26)/annual dose. It was ETB 75.23 (95% CI: 72. 97–74.49) for market oriented farmers and ETB 42.6 (95%CI: 41.24–43.96) for mixed crop livestock farmers. Willingness to pay for the vaccine was significantly higher for farmers in market oriented system than in mixed crop livestock system. It was also significantly higher for farmers whose main livelihood is livestock than those whose main livelihood is other than livestock, and for farmers who keep exotic breed cattle and their crosses than those who keep only local cattle breeds. Willingness to pay significantly increased with increase in FMD impact perception and vaccine knowledge scores of farmers. The high mean WTP estimates showed that farmers are enthusiastic about using the FMD vaccine. Market-oriented farmers with higher willingness to pay may be more likely to pay full cost if official FMD vaccination is planned in the country than mixed crop livestock farmers. Animal health extension about livestock diseases impact and vaccines has a potential to increase farmers' uptake of vaccines for disease control.

**Funding:** "This study was funded by University of Gondar (www.uog.edu.et). The grant was awarded to WTJ (the first author of this article) with grant letter number VP/RCS/05/280/2016. The funders had no role in study design, data collection and analysis, decision to publish, or preparation of the manuscript."

**Competing interests:** The authors have declared that no competing interests exist.

## 1. Introduction

Foot and mouth disease (FMD) is arguably the most important disease of livestock worldwide due to costs associated with production losses, trade restriction, and prevention and control [1]. The disease is caused by foot and mouth disease virus of the genus *Aphthovirus* and family *Picornaviridae*, and primarily affects cattle, swine and small ruminants [2].

Historically two major approaches have been used separately or in combination to control FMD worldwide: intensive surveillance and stamping out as in case of United Kingdom, Scandinavia and North America, and vaccination with or without stamping out as in the case of continental Europe and parts of South America [3]. Currently, the stamping out strategy is used by disease free developed countries to control incursion of outbreaks through rapid detection of disease introduction and slaughtering of infected and in contact herds. Regular mass vaccination is often used to control the disease in endemic developing countries.

For endemic developing countries, prophylactic vaccination remains the main feasible method of controlling the disease [4]. However, FMD vaccination is complex as compared to other similar epidemic livestock diseases such as rinderpest and peste des petitis ruminants, which have effective and affordable vaccines. Conventional FMD vaccines are inactivated vaccines and need frequent application. Vaccine matching is also a serious challenge in FMD vaccination due to existence of multiple serotypes and strains that don't or poorly cross-protect against each other [5, 6]. Foot and mouth disease vaccines are often made to cover multiple serotypes and strains and these negatively affect the potency and cost of the vaccines as compared to monovalent vaccines [7]. This makes control of FMD using vaccination a difficult undertaking for resource constrained developing countries like Ethiopia.

Although foot and mouth disease is endemic in Ethiopia, control of the disease using vaccination has rarely been practiced. Despite the stated interest and plan of the government to improve the disease situation and boost meat and live animal export trade [8], no official control has been practiced yet. Unlike other transboundary livestock disease such as peste des petitis ruminants, lumpy skin disease, contagious bovine pleuropneumonia, African horse sickness, sheep and goat pox among others, for which the government is providing free vaccination service for farmers, no support have been given for FMD vaccination. Except some market oriented farmers in urban and periurban areas, most farmers are not vaccinating their herds against FMD. The vaccine is not adequately available in the market and the main reason for this could be the perception that FMD vaccines are expensive and farmers may be reluctant to use the vaccine. The present study was conducted to assess farmers' willingness to pay (WTP) for a quality FMD vaccine and identify the factors that could potentially influence their WTP using contingent valuation method in Amhara region of Ethiopia.

## 2. Materials and methods

### 2.1 Ethics statement

The study was ethically reviewed and approved by Institutional Review Board of University of Gondar. Oral informed consent has been obtained from questionnaire respondents.

### 2.2 The study area and population

The study was conducted in the Gondar-Bahir Dar milk shed in Amhara region of Ethiopia. This area encompasses livestock producers who supply milk to the two major cities of northwestern part of the Amhara regional state namely Bahir Dar and Gondar [9]. Broadly there are two types of production systems practiced in the region: the dominant mixed crop-livestock (MCL) production system, which is a subsistence system that is practiced in the rural areas,

and a market oriented (MO) production system which produces commercial milk and is practiced in urban and periurban areas.

## 2.3 Contingent valuation method

There are many varieties of techniques used for valuation in economics, grouped in two categories: revealed preference and stated preference. Revealed preference techniques are based on actual behavior of individuals in a real market reflecting utility maximization subject to constraints. Stated preference techniques, on the other hand, are based on responses of individuals to hypothetical questions rather than from observations of real-world choices [10]. As the responses are contingent upon the specific conditions laid out in the hypothetical market, this form of stated preference methods are broadly referred to as contingent valuation [10]. Contingent valuation method (CVM) is a widely used nonmarket valuation method in economics to determine WTP for goods or services that are not traded in the market place. This method of measuring value is developed and widely used in environmental economics where it is used to value environmental amenities and services [10, 11]. The other areas in economics where this method is increasingly being applied are health, transportation safety, and cultural economics [11].

In CVM, a survey is designed to create different hypothetical market scenarios for reflecting value of non-marketable goods and survey respondents are asked to state their response to the hypothetical market scenarios. The data collected by such surveys is then analyzed in a similar manner as the choices made by consumers in actual markets [12]. Despite the controversy over the validity of this method of valuation, it is a popular nonmarket valuation method in environmental economics [11].

Willingness to pay studies using contingent valuation methods are also increasingly being used in in animal disease control in recent years [13–17]. Brief review of literature on earlier application of contingent valuation method in animal health and associated areas can be found in Bennet and Balcombe [13]. Although animal disease vaccines are marketable and do have market prices, they also have public good nature. Hence, contingent valuation can be used to determine the WTP for these public goods. This could be for a vaccine under development and yet not marketed [13] or for existing vaccines, which are poorly adopted for variety of reasons including price sensitivity [14, 15, 17].

There are different methods of WTP data elicitation (bid design) in CVM. Possible bidding mechanisms include: bidding, payment card, Open-ended question, and Single (Double)-bounded dichotomous choice methods [18]. Dichotomous choice methods are important in that they have less starting bias and simplicity and are therefore commonly used methods in contingent evaluation. While the double dichotomous choice is more complex analytically, it has an advantage in the data efficiency [19] and hence has been a choice of method for the present study.

## 2.4 The survey

The survey was prepared in line with the recommended contingent evaluation elicitation guideline of the National Oceanic and Atmospheric Administration [18]. It consisted of two major parts. The first part contained the bidding questions directly related to the WTP for a hypothetical vaccine market and the second part contained the questions about the socioeconomic factors that could influence the willingness of the farmer to pay for FMD vaccine (S1 File). The first question of the survey before the two major parts was a question verifying whether the respondents knew the disease. The bidding questions consisted of double-bound dichotomous choice questions. In the double dichotomous choice bidding format, there were questions on two stages. The initial stage questions contain a set of bid amounts to which

respondents state their WTP as 'yes' or 'no' to hypothetical vaccine prices. These initial bid amounts were distributed equally and randomly among respondents during questionnaire administration. The initial stage questions were followed up by second stage questions with bid amount of 50% plus or minus to the initial bid amounts depending on the response to the first bid amount. If the response to the initial bid amount was 'yes' the follow up bid amount would increase by 50% and if the response to the initial bid amount was 'no' the follow up bid amount would decrease by 50% (Table 1). In both the initial and follow up stage questions an 'undetermined' alternative was included for respondents who were not able to decide as 'yes' or 'no'. The hypothetical market scenario was followed by a debriefing question to ensure that the respondents correctly understood the presented scenario before they gave their answers.

The initial bid set contained prices of Ethiopian Birr (ETB*, One Ethiopia Birr is equivalent to 0.034 USD at the time of the survey.) 20, 40, 60 and 80 per dose. This price set was proposed based on information found from open ended WTP pilot survey on 15 MCL and 15 market oriented farmers with price range of ETB 5-100/annual dose for the same hypothetical vaccine. This price range was roughly similar to the range of USD 0.4–3 (ETB 12–88) for different types of FMD vaccines reported in the literature in different countries in the world [1, 20]. Since, a quality standard FMD vaccine should have a protection level greater than 75% [21], in this study the respondents were made to bid for a hypothetical vaccine with protection effectiveness of 80% which is proposed to be administered twice a year by the public veterinary service in their village.

The second part of the questionnaire contains questions related to sociodemographic features and husbandry practices of the respondents that could potentially affect the respondent's WTP for FMD vaccine. These include demographic variables (Age, Educational status, Main livelihood), livestock husbandry related variables (number of cattle owned, number of TLU owned, income from cattle sale, income from milk sale, cattle kept for business, breed of cattle kept, main veterinary service used, experience of vaccine for livestock), perception about FMD impact on livestock, and knowledge on the use of vaccine for livestock disease prevention (S1 File). The perception of FMD impact and the knowledge of vaccine variables were measured as composite scores of several questions under each variable. The FMD impact perception score was generated from five questions each of which has a maximum score of three (giving a total maximum impact score of 15). Similarly the knowledge score about livestock vaccine was generated based on four dichotomous vaccine knowledge questions which have score of either one (correct answer) or zero (incorrect answer) giving a maximum knowledge score of four (S1 File).

The survey was administered by means of face to face interviews of respondents in the local language (Amharic) and was done by trained veterinary personnel in each district. The study protocol was ethically reviewed and approved by Institutional Board of University of Gondar. The respondents were asked for their informed consent before the interview.

**Table 1. The double-bound dichotomous choice questionnaire bid structure.**

| Initial bid (administered randomly one for each respondent) (ETB) | Initial bid response | Follow up bid amount (ETB) |
|---|---|---|
| 20 | no | 10 |
| | yes | 30 |
| 40 | no | 20 |
| | yes | 60 |
| 60 | no | 30 |
| | yes | 90 |
| 80 | no | 40 |
| | yes | 120 |

## 2.5 Sampling and sample size

The respondents for the WTP survey were farmers from four districts in Bahir Dar–Gondar Milk shed in northwest Ethiopia that represent MO and MCL production systems. The two urban districts (Bahir Dar and Gondar) represent the MO and the other two rural districts (Gondar Zuria and Estie) represent MCL system. These four districts were selected purposefully and conveniently to represent the two types of livestock production systems in the area. While the two urban districts are the major urban centers in the milk shed and had recent FMD history, the two rural districts were selected because of recent history of FMD outbreaks. Districts with a recent history of FMD outbreak were particularly considered for sampling to get more farmers who are familiar with the disease to respond to the survey questions. A total of 400 farmers (farm household heads), 100 from each district who knew the disease, were enrolled for the survey. The farmers were sampled at some haphazard interval in different streets of the urban districts and villages in the rural districts. Strict randomization of the selection was not possible due to lack of sampling frame and in accessibility of some of the villages in the rural districts.

## 2.6 Data analysis

Interval regression analysis [22] was used to estimate the farmers' WTP for FMD vaccine from the double-bounded dichotomous contingent valuation data collected with the questionnaire.

The responses to the double-bounded CV questions give four possible discrete outcomes:1) the respondent was not willing to purchase the FMD vaccine both at initial bid amount and at the lower follow up bid amount ("no", "no"); 2) the respondent was not willing to purchase the FMD vaccine at the initial bid amount but was willing to buy at the lower follow up bid amount ("no", "yes"); 3) the respondent was willing to purchase FMD vaccines at the initial bid amount but not at the higher follow up bid amount ("yes", "no"); or (4) the respondent was willing to purchase the FMD vaccine at both the initial bid amount and the higher follow up bid amount ("yes", "yes"). This creates four possible intervals where farmers WTP could fall: $(0, B_l)$, $(B_l, B_i)$, $(B_i, B_h)$, $(B_h, \infty)$. Where, $B_i$ the initial bid amount, $B_l$ is the lower follow up bid amount, and $B_h$ is the higher follow up bid amount. This results in three types of censoring: left censored, right censured and interval censored. The WTP was modelled from the interval data created this way and interval regression was used to estimate the mean WTP and potential factors that influence the WTP amount (S1 Dataset). In the model, WTP is estimated as linear function of respondents' characteristics with normal distribution of random error [23].

The data was first entered into Microsoft excel for editing and cleaning and then taken to Stata software version 14 (Stata Corp. College Station, TX) for analysis. In the interval regression modelling, first the variables considered for the models were checked for multicollinearity using variance inflation factor (VIF). Variance inflation factor value of above 10 was considered as indicator of presence of collinearity [24]. Then the full models containing all non collinear variables were run. Final models were reached through backward elimination of non-significant variables (p-value > 5%) one at a time until only significant variables were left.

## 3. Results

### 3.1 Sociodemographic and cattle husbandry characteristics of the survey respondents

A total of 398 respondents participated in the survey. After cleaning, the data from 386 respondents were used for all the analyses. Data of 13 respondents were excluded from the analyses due to incomplete or inconsistent responses for one or more important variables. Data of

three respondents were excluded from the WTP analysis because they gave 'undetermined' response for WTP questions.

Sociodemographic and cattle husbandry characteristics of the survey respondents are summarized in Table 2. Livestock was the main livelihood for majority (58%) of MO respondents but only very few MCL respondents stated livestock as main livelihood source. The total numbers of different species of livestock kept by respondents were aggregated using tropical livestock units (TLU) and the average TLU holding was about six TLUs being a little bit higher for MCL than MO respondents. Almost all respondents in the MO system keep cattle for business (sale of cattle or milk or other products) whereas only about 28% of MCL respondents keep cattle for business. Majority of the respondents (85%) use modern veterinary service and 88% have experience of using vaccine in their cattle husbandry. Unexpectedly, the MO respondents

**Table 2. Summary of sociodemographic and livestock husbandry characteristics of survey respondents by production system (MCL = 195, MO = 191, Overall = 386).**

| Variables | MCL (mean (SD[a])) | MO (mean (SD)) | Overall (mean (SD)) |
|---|---|---|---|
| **Sex** | | | |
| Male | 0.98 (0.14) | 0.91 (0.29) | 0.94 (0.23) |
| Female | 0.02 (0.14) | 0.08 (0.29) | 0.06 (0.23) |
| **Age** | 47.1 (9.2) | 44.0 (10) | 45.6 (9.7) |
| **Educational status** | | | |
| Illiterate | 0.60 (0.49) | 0.22 (0.42) | 0.41 (0.50) |
| Primary | 0.36 (0.48) | 0.61 (0.49) | 0.48 (0.50) |
| Secondary and above | 0.03 (0.17) | 0.17 (0.37) | 0.10 (0.30) |
| **Main livelihood** | | | |
| Livestock | 0.02 (0.12) | 0.58 (0.49) | 0.30 (0.45) |
| Other (crop, trade, employment) | 0.98 (0.12) | 0.42 (0.49) | 0.70 (0.45) |
| **Number of cattle owned** | 6.3 (3.1) | 7.4 (5.3) | 6.9 (4.4) |
| **Number of TLU** | 6.6 (3.2) | 5.3 (3.7) | 6.0 (3.5) |
| **Income from cattle sale for the previous year** | 5838 (8174) | 22632 (26927) | 14148 (21498) |
| **Income from milk sale for the previous year** | 438 (2812) | 51425 (62420) | 25667 (50777) |
| **Cattle kept for business** | | | |
| yes | 0.29 (0.46) | 1 (0) | 0.64 (0.48) |
| no | 0.71 (0.46) | 0 (0) | 0.34 (0.48) |
| **Breed of cattle kept** | | | |
| Local | 0.65(0.48) | 0.08 (0.28) | 0.37 (0.48) |
| Exotic and their cross | 0.03 (0.17) | 0.31 (0.46) | 0.17 (0.38) |
| Both local, exotic and their cross | 0.32 (0.47) | 0.60 (0.49) | 0.46 (0.50) |
| **Main veterinary service used** | | | |
| Traditional | 0 | ~0 | ~0 |
| modern | 0.91 (0.28) | 0.79 (0.41) | 0.85 (0.36) |
| Both | 0.09(0.28 | 0.21 (0.41) | 0.15 (0. 36) |
| **Ever used vaccine for livestock** | | | |
| Yes | 0.98 (0.12) | 0.78 (0.42) | 0.88 (0.32) |
| no | 0.02 (0.12) | 0.22 (0.42) | 0.12 (0.32) |
| **Perception of FMD impact (out of 15 score scale)** | 9.9 (3.1) | 10.7 (1.6) | 10.3 (1.5) |
| **Knowledge about livestock vaccine (out of 4 score scale)** | 3 .4 (0.8) | 2.7 (0.9) | 3 (0.1) |

[a]SD = standard deviation

**Table 3. Summary of WTP responses in the double dichotomous contingent evaluation survey (N = 393).**

| Initial bid in ETB | Initial bid response | | Follow up bid amount and response | | |
|---|---|---|---|---|---|
| | Response | No. of response (%) | Follow up bid in ETB | No. of 'no' responses (%) | No. of 'yes' responses (%) |
| 20 | no | 14 (16) | 10 | 2 (14) | 12 (86) |
| | yes | 82 (84) | 30 | 29 (35) | 53 (65) |
| 40 | no | 32 (34) | 20 | 14 (44) | 18 (56) |
| | yes | 63 (66) | 60 | 29 (46) | 34 (54) |
| 60 | no | 50 (50) | 30 | 25 (50) | 25 (50) |
| | yes | 50 (50) | 90 | 24 (48) | 26 (52) |
| 80 | no | 62 (67) | 40 | 29 (47) | 33 (53) |
| | yes | 30 (33) | 120 | 14 (47) | 16 (53) |
| **Overall** | **No** | **158 (0.41)** | | **70 (47)** | **88 (53)** |
| | **Yes** | **225 (0.59)** | | **96 (43)** | **129.(57)** |

use traditional veterinary service more and have less experience with livestock vaccine as compared to the MCL respondents (Table 2).

The average score of the overall FMD impact perception was 10.3 out of a total of 15 points, which was a little bit higher for the urban MO system respondents. Similarly, the average vaccine knowledge score was 3 out of total 4 points, which was higher for MCL respondents (Table 2).

## 3.2 Willingness to pay for FMD vaccine and factors affecting WTP

The willingness to pay for the hypothetical vaccine presented was observed to decrease with an increase in the bid amount (Table 3). The percentage of WTP ('yes' responses) for the initial bids of 20 birr, 40 birr, 60 birr and 80 birr were 84%, 66%, 50% and 33% respectively.

Interval regression analysis showed that the mean WTP as determined by the constants of the null models (a model without any explanatory variables) was ETB 57.76 (95% CI: 53.74%-61.78) per dose for all respondents, ETB 42.66 (95%CI: 38.32–46.99) for MCL respondents and ETB 74.56 (95% CI: 67.91–81.24) for MO respondents. These are WTP estimates without taking into account any variables that could potentially affect WTP for the vaccine.

None of the sociodemographic variables considered in the overall interval regression model were significantly associated with WTP (P> 0.05). The variables found to be significantly associated with WTP were those related to livestock husbandry which include livestock production system, type of cattle breeds kept, whether livestock is main livelihood or not, perception of impact of FMD, and knowledge about livestock vaccines (Table 4). For example, the WTP in

**Table 4. Livestock husbandry variables significantly associated with WTP for FMD vaccines (N = 383).**

| Variables | | Model Coefficients | 95% CI of Coefficients | P- value |
|---|---|---|---|---|
| **Production system** | MO | 18.92 | 8.69–29.14 | <0.001 |
| | MCL | reference | reference | reference |
| **Main livelihood** | livestock | 15.68 | 5.63–25.76 | 0.002 |
| | Other | reference | reference | reference |
| **Breed of cattle kept** | | | | |
| | Exotic and their cross | 13.56 | 0.59–26.52 | 0.040 |
| | Both local, exotic and their cross | -1.10 | -9.82–7.61 | 0.804 |
| | local | reference | reference | reference |
| **Perception of FMD impact** | | 4.58 | 2.15–7.00 | <0.001 |
| **Knowledge about livestock vaccine** | | 6.70 | 2.80–10.60 | 0.001 |

**Table 5. Livestock husbandry variables significantly associated with WTP for respondents in the different production systems (N = 191 for MCL and N = 192 for MO).**

| Production systems and Variables | | Model Coefficients | 95% CI of Coefficients | P- value |
|---|---|---|---|---|
| **CLM system** | | | | |
| No of cattle owned | | 1.61 | 0.25–2.96 | 0.020 |
| Perception of FMD impact | | 4.20 | 1.15–7.25 | 0.007 |
| Knowledge about livestock vaccine | | 6.06 | 1.18–10.93 | 0.015 |
| **MO system** | | | | |
| Main livelihood | livestock | 14.95 | 2.92–26,98 | 0.015 |
| | other | Reference | reference | reference |
| Perception of FMD impact | | 5.96 | 2.00–9.92 | 0.003 |
| Knowledge about livestock vaccine | | 8.09 | 1.70–14.47 | 0.013 |

MO farmers is 18.92 ETB higher than the MCL farmers, and when the vaccine knowledge score of farmers increases by one unit the WTP increases by 6.7 ETB, keeping other variables in the model constant.

Given the socioeconomic and husbandry difference between MCL and MO production systems and also the significant difference in WTP for an FMD vaccine, separate interval regression was run for the two production systems and the results are shown in Table 5.

For the MCL respondents, the factors that significantly influence the WTP of FMD vaccine were number of cattle owned, perception of FMD impact and Knowledge of livestock vaccine in which increase of the value in all of them increases WTP (Table 5). For the MO respondents, factors that were significantly associated with WTP of FMD vaccine were whether livestock is main livelihood or not, and FMD perception score and livestock vaccine knowledge scores. Having livestock as main livelihood, higher livestock impact perception score and higher livestock vaccine knowledge scores significantly increased WTP.

The mean and median WTP estimates were also derived (Table 6) from the interval regression models described in the preceding paragraphs. The WTP derived from the model are very close to the estimates directly observed from the intercept of the null models described earlier in this section.

## 4. Discussion

### 4.1 Willingness to pay

Despite the availability of an FMD vaccine, the use of such a vaccine in Ethiopia, especially in the dominant subsistence livestock productions (pastoral system and mixed crop livestock system) has been rare. This is presumably due to the low availability and high cost of FMD vaccine. In this study, we tried to estimate how much farmers in two typical Ethiopian livestock production systems in the Bahir Dar-Gondar Milk shed are willing to pay for a quality FMD vaccine and what sociodemographic and livestock husbandry characteristics influence farmers' WTP for the vaccine. The study revealed that majority of the farmers answered 'yes' to both

**Table 6. WTP estimates derived from the best interval linear regression models.**

| Group | Mean | Median | Standard deviation | 95% confidence interval |
|---|---|---|---|---|
| MCL system | 42.60 | 42.84 | 9.55 | 41. 24–43.96 |
| MO system | 75.23 | 73.34 | 15.91 | 72. 97–74.49 |
| Overall | 58. 23 | 51 .80 | 20.24 | 56.20–60.26 |

the initial and follow up price bids set for FMD vaccine indicating their enthusiasm for using the vaccine. The proportion of farmers willing to pay decreases monotonically from 82% to 32% when the initial bid values increased from ETB 20 to 80, which is consistent with economic law of demand [25]. This pattern assures the rationality and hence the validity of the responses given by the farmers.

The estimates of the WTP for the vaccine reported from the study were generally high; for example, it is much higher than the ETB 20/annual dose currently available in the government vaccine production institute in the country. This is unexpected in a region where farmers get most of the livestock vaccines with substantially cheaper prices or even free as in the case of vaccines for transboundary animal diseases other than FMD. Although reason for willingness was not asked in the survey, farmers were unpromptingly explaining that if the disease occurs its impact on milk reduction will be very high compared to the stated vaccine prices.

The mean WTP for the vaccine as estimated using the interval regression model parametrized from the double-bound dichotomous questionnaire data was ETB 58.23 (USD1.96) per year. The WTP was significantly different in the different production systems. As expected, it was higher (ETB 75.23 (USD 2.53)) for the MO and lower (ETB 42.6 (USD 1.43)) for mixed crop-livestock system. These WTP estimates are much higher than the ETB 20 (ETB10/dose for a biannual vaccination) currently charged for the trivalent (O, A, SAT2) vaccine produced by the National Veterinary Institute in the country. However, the provision of the vaccine at this price by the institute may not reflect the real market value as the government usually provides vaccines at subsidized prices. Moreover, availability of this vaccine is limited and its effectiveness has also been in question (pres. communication). Therefore, it is difficult to assert that farmers are willing to pay more than market price based on this comparison. The estimates were within the range of USD 0.4 to 3 cost paid per dose for FMD vaccine including vaccine delivery across the world [1]. In Tanzania a roughly similar WTP amount, i.e. USD 1.84 (95% CI: 1.28–2.48) was reported for cattle FMD vaccine [16]. An FMD economic impact study in traditional smallholder production system in Ethiopia indicated the potential economic profitability of FMD vaccination using likely market price of FMD vaccine [26]. The observed farmers' WTP for the vaccine can, therefore, be considered as economically justifiable. Generally, the average WTP stated by the farmers indicated that they are willing to pay substantial amount, if a quality vaccine is presented and its use promoted. This could be, for example, full cost coverage for the market oriented farmers and the substantial part of the price for the dominantly subsistence MCL system. However, several studies on potential biases associated with WTP determined in contingent evaluation consistently indicated that it tends to overestimate the WTP as compared to actual market behavior [27–29]. This has to be taken into consideration when the estimated WTP are interpreted for practical application.

## 4.2 Factors affecting willingness to pay

A number of sociodemographic and cattle husbandry variables were evaluated for their influence on WTP of the farmers. None of the sociodemographic variables considered such as sex, age, household size, education status and livestock number (proxy for income level) had significant impact on WTP. However, it was observed that MCL farmers who have relatively lower level of education status had more experience of using livestock vaccine than MO farmers who had better education status. Hence, it seems that the main driver for vaccine awareness and uptake is not related to formal education level. Probably the access to livestock extension, which is better in the rural MCL system, might play greater role for better uptake of vaccine in this system.

Livestock husbandry related variables such as livestock production system, type of cattle breeds kept, whether livestock is the main livelihood or not, perception of impact of FMD and

knowledge about livestock vaccines were found to be important drivers of WTP for FMD vaccine. Willingness to pay was significantly higher for MO than in the MCL system. Farmers in the urban market oriented system keep more productive but disease susceptible animals for market milk production, hence, would be sensitive to impact of FMD and it is economically rational that they were willing to pay more for the vaccine. In support of this an FMD economic impact study in the same areas indicated much higher loss of USD 459 (USD 26 per animal in the affected herd) due to FMD outbreak in a MO farms as compared to USD 34 (USD 5 per animal in affected herd) in CLM farms [30]. Willingness to pay was also higher for respondents whose main livelihood is livestock raising than respondents whose main livelihood is other than livestock. This is logically consistent, as farmers would like to safeguard their source of livelihood by paying more for vaccination. Respondents who keep exotic breed cattle and their crosses showed higher WTP than those who keep only local cattle breeds. This could be related to the factors discussed in the preceding sentences. Market oriented farmers whose main livelihood is driven from dairy production would keep more exotic cross bred cattle than their counterparts and their higher WTP for FMD can be similarly explained. At this point one might pose question of multicollinearity among these factors i.e. production system, source of livelihood and breed for cattle kept. But their multicollinearity was tested during the model building and no multicollinearity was found between any of the variables large enough to drop from the analysis.

It was also observed that WTP increases significantly with increase in FMD impact perception score. Similarly, farmers with high risk perception for bovine tuberculosis (BTB) in UK were seen to have higher WTP for BTB vaccine [13]. It is economically rational that farmers who perceived significant impact of the disease are willing to pay more to avoid the disease.

Higher livestock vaccine knowledge score was significantly positively associated with WTP. Those who had better knowledge were observed to be willing to pay more. Similar finding of increasing WTP with increase in vaccine knowledge level was reported for other livestock vaccines [17]. This has been also observed in human vaccine where people who have better vaccine related knowledge are willing to pay more [31]. This indicates there is a room for increasing WTP and uptake of vaccine by increasing vaccine awareness related extension to farmers for control and eradication of livestock diseases.

## 5. Conclusions

The farmers' mean WTP for FMD vaccine in the study area was generally found high and was greater than the price of the vaccine currently produced and sold by the national veterinary institute in the country. The study findings contested the perception that FMD vaccines are costly and farmers would be reluctant to pay for it. The estimated WTP prices, especially that of the market-oriented farmers can be within the range of FMD vaccine price available in the world market. Based on these WTP estimates it can be assumed that market-oriented farmers with higher willingness to pay may be more likely to pay full cost if official FMD vaccination is planned in the country than mixed crop livestock farmers. Farmers who have high perception of FMD risk and good knowledge of vaccines have greater WTP for FMD vaccine. Hence, animal extension service about the disease impact and importance of vaccines in livestock disease control has potential to increase farmers' uptake of vaccines for disease control.

## Supporting information

**S1 File. Willingness to pay questionnaire survey.**
(DOCX)

**S1 Dataset.**
(XLSX)

## Acknowledgments

Authors would like thank the farmers who were willing to participate as respondents for the survey.

## Author Contributions

**Conceptualization:** Wudu T. Jemberu, Wassie Molla.

**Data curation:** Wudu T. Jemberu, Wassie Molla, Tigabu Dagnew.

**Formal analysis:** Wudu T. Jemberu, Wassie Molla, Tigabu Dagnew.

**Funding acquisition:** Wudu T. Jemberu.

**Investigation:** Wudu T. Jemberu, Tigabu Dagnew, Henk Hogeveen.

**Methodology:** Wudu T. Jemberu, Wassie Molla, Tigabu Dagnew, Jonathan Rushton.

**Project administration:** Wudu T. Jemberu.

**Supervision:** Jonathan Rushton, Henk Hogeveen.

**Validation:** Wassie Molla, Jonathan Rushton, Henk Hogeveen.

**Visualization:** Wudu T. Jemberu, Henk Hogeveen.

**Writing – original draft:** Wudu T. Jemberu, Wassie Molla.

**Writing – review & editing:** Tigabu Dagnew, Jonathan Rushton, Henk Hogeveen.

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
