## [Decision Letter · Decision Letter 0]

8 Jul 2020

PONE-D-20-17667

Farmers' willingness to pay for foot and mouth disease vaccine in different cattle production systems in Amhara region of Ethiopia.

PLOS ONE

Dear Dr. Jemberu

Thank you for submitting your manuscript to PLOS ONE. After careful consideration, we feel that it has merit but does not fully meet PLOS ONE’s publication criteria as it currently stands. Therefore, we invite you to submit a revised version of the manuscript that addresses the points raised during the review process.

Many thanks for submitting your manuscript to PLOS One

Your manuscript was reviewed by two experts in the field, and they have recommended some modifications be made prior to acceptance

If you could write a response to reviewers, then that will expedite reviews upon resubmission

I wish you the best of luck with your revisions

Hope you are keeping safe and well in these difficult times

Thanks

Simon

We look forward to receiving your revised manuscript.

Kind regards,

Simon Clegg, PhD

Academic Editor

PLOS ONE

2. Please include additional information regarding the survey or questionnaire used in the study and ensure that you have provided sufficient details that others could replicate the analyses.

For instance, if you developed a questionnaire as part of this study and it is not under a copyright more restrictive than CC-BY, please include a copy, in both the original language and English, as Supporting Information.

3. In your Methods section, please provide additional information about the participant recruitment method and the demographic details of your participants.

Please ensure you have provided sufficient details to replicate the analyses such as:

a) the recruitment date range (month and year),

b) a description of any inclusion/exclusion criteria that were applied to participant recruitment,

c) a table of relevant demographic details,

d) a statement as to whether your sample can be considered representative of a larger population,

e) a description of how participants were recruited, and

f) descriptions of where participants were recruited and where the research took place.

Reviewers' comments:

Reviewer's Responses to Questions

**Comments to the Author**

1. Is the manuscript technically sound, and do the data support the conclusions?

Reviewer #1: Partly

Reviewer #2: Yes

2. Has the statistical analysis been performed appropriately and rigorously? 

Reviewer #1: No

Reviewer #2: Yes

3. Have the authors made all data underlying the findings in their manuscript fully available?

Reviewer #1: Yes

Reviewer #2: Yes

4. Is the manuscript presented in an intelligible fashion and written in standard English?

Reviewer #1: Yes

Reviewer #2: Yes

5. Review Comments to the Author

Reviewer #1: The study is an important one as it can lead to better control of FMD in Ethiopia. Though the appropriate data has been collected and most of the analyses done, the analysis mentions procedures like checking for multicollinearity but the details of how are missing. It is not clear how WTPs were constructed in the analysis. The resultant coefficients in the regression analysis are not interpreted at all, only the p values (significance).

Specifically:

L24: Replace ‘rarely’ with ‘sparingly’and expensiveness with 'high cost'

L34-37: There are too many variables in this sentence. One is unable to see what is being compared. Split the sentence to make clear what WTP is significantly higher than which with regard to systems, livelihoods, breeds

L40: Report in the affirmative rather than in the negative by turning the sentence around

L42: market-oriented system farmers

L43-44: Be specific on which vaccine, which disease because your study was not on all vaccines, all diseases

L56: Write UK in full; Scandinavia

L65: Do you mean endemic rather than epidemic?

L67: Replace ‘vaccination’ with ‘application’ otherwise it reads like the vaccines are being vaccinated

L68: ‘negatively affect the effectiveness and raise the cost of vaccination’

L75: Sheep and goat

L77: replace etc. with among others

L78: vaccine not vaccines

L149: What does it mean ‘undetermined alternative’

The survey should really be in reverse. You first get the perception and knowledge of the farmers on FMD as you need them as independent variables. Then you need to make all of them understand the disease and its control by the vaccine whose attributes they need to understand before bidding because as per your background information, these farmers may not even know the vaccine. Then you pose the bidding questions. Apparently before bidding you did not explain these aspects to the farmers which makes one wonder whether they knew what they were bidding for.

L217: Change check to checked; It is not clear how multicollinearity was checked

It is rare that a backward elimination of non-significant variables will result in a model with only significant variables

L226: What does ‘inconsistent response’ mean?

L227: Who gave the ‘undetermined response’ for WTP?

L230: The ‘majority’ here is only about half. How were the results of the other half. Explain from the table without repetition

Table 2: In the column, align contents conventionally; text aligned to the left and figures to the right. What is income from cattle and milk sale? Annual, monthly or what?

L255-260: It is not clear how the mean WTPs were calculated

L262: The p should be >0.05 if no significant association

Table 4: Just indicate the reference category within the table by writing ref in the model coefficients column for the respective reference categories instead of having a footnote which is not clear

L261-265 and 272-278: Give the interpretation of the model coefficients, not just of the p value (significance). Considering the regression equation that arises from the results would help

L272-276: Remove unnecessary capitalization of words

Table 6: It is not clear how the WTP estimates were derived

L300: citation/reference?

L311: What is ‘parametrized’?

L321-322: Correct the sentence

L328: What is ‘quality vaccine’ – you did not explain the vaccine to them

L330: What is CLM?

L332: Change ‘tend’ to ‘tends’

L348: Avoid an abbreviation at the beginning of a sentence

L362-365: Sentence is too long and needs rephrasing for clarity

L374: Correct the word knowledge

In the conclusion, WTP alone is not sufficient to make the conclusion that vaccine price can be covered by farmers fully without your knowledge on their ‘ability to pay’. You need to look at their profits or how they perceive profitability from their enterprises. As is, it is possible that WTP may represent valuation of vaccination rather than actual desire to pay the amounts

L441: correct 20.20

Reviewer #2: Reviewer comments:

Excellent study design, data analysis, and overall manuscript. A few suggestions below. Suggest one more read through by a strong English speaker to remove typos. Identified some below, but there are probably more, and they distract from the quality of the paper.

Line 35-36: should read “whose main livelihood is”

36 “livestock than those whose main livelihood is other than livestock,” – Rewrite, confusing

65 – typo – should read peste

63 onwards – May be worthwhile to mention the role of diagnostic testing in FMD vaccination as an additional challenge – alluded to but not directly mentioned

78 – should read “vaccine is…”

120 – contingent valuation is also appropriate in this situation because many people are not vaccinating.

180 – should read “asked for”

198 – should read farmers’

199 – should read double-bounded

347 – should read “is the main”

370 – should read “are willing to pay more”

373 – should read “pay more”

381 – should read farmers’

385 – should read farmers

Gender – noticed the percentage of women respondents is very low. Do you think this is representative of the study population? Seems possible the selection of participants may bias against women if they are more likely to be at home. May be worthwhile to think more about this in the discussion.

6. PLOS authors have the option to publish the peer review history of their article (what does this mean?). If published, this will include your full peer review and any attached files.

Reviewer #1: No

Reviewer #2: No

---

## [Author Response · Author response to Decision Letter 0]

18 Aug 2020

Responses to reviewers’ comments 

Note: The line number refer to the Revised Manuscript with Track Changes 

Reviewer #1: 

Comment: The study is an important one as it can lead to better control of FMD in Ethiopia. Though the appropriate data has been collected and most of the analyses done, the analysis mentions procedures like checking for multicollinearity but the details of how are missing. It is not clear how WTPs were constructed in the analysis. The resultant coefficients in the regression analysis are not interpreted at all, only the p values (significance).

Response: 

- Detail of how multicollinearity was assessed for covariates in the model is provided in the revised version (L219-22)

- How the willingness to pay was derived from interval regression model was documented in L 212- 215 of the manuscript. It is explained in more detail below in the responses given for the specific comments on the issue. 

- The interpretation of the model coefficients are provided in the revised version (L269-272) 

Specifically:

L24: Replace ‘rarely’ with ‘sparingly’ and expensiveness with 'high cost'

Response: modified as suggested (L24)L34-37: There are too many variables in this sentence. One is unable to see what is being compared. Split the sentence to make clear what WTP is significantly higher than which with regard to systems, livelihoods, breeds

Response: The sentence split as suggested for easy reading (L34-37)

L40: Report in the affirmative rather than in the negative by turning the sentence around

Response: the sentence changed as suggested (L40-41)

L42: market-oriented system farmers

Response: modified as suggested (L 44)

L43-44: Be specific on which vaccine, which disease because your study was not on all vaccines, all diseases

Response: The vaccine knowledge was asked for livestock vaccine in general as farmers didn’t vaccinate against FMD and have no experience with FMD vaccine to give FMD vaccine specific information. This was stated in the methodology L177 and the appendix)

L56: Write UK in full; Scandinavia

Response: corrected as suggested (L 57)

L65: Do you mean endemic rather than epidemic?

Response: this was to refer to the epidemic nature of the diseases (FMD, PPR, LSD etc) which always occur in the form of outbreak instead of more or less constant number of cases throughout a year as endemic disease like mastitis or bovine paratuberculosis etc. 

L67: Replace ‘vaccination’ with ‘application’ otherwise it reads like the vaccines are being vaccinated

Response: changed as suggested (L68)

L68: ‘negatively affect the effectiveness and raise the cost of vaccination’

Response. This was as compared to monovalent vaccines. When multiple strains are included in the vaccine, the host immune system exhaust and produce less immune response than when antigens are injected separately and hence reduce potency, and at the same time the cost increases because of inclusion of antigen load for each serotype. The sentences is modified to make this clear (L69-70) 

L 75: Sheep and goat

Response: Corrected (L77)

L77: replace etc. with among others

Response: modified as suggested (L78)

L78: vaccine not vaccines 

Response: corrected (L80)

L149: What does it mean ‘undetermined alternative’

Response: This is an option given to respondents when they could not decide ‘yes’ (agree) or ‘no’ (disagree) with proposed vaccine price. Including this option of “’undetermined or “no answer” is customary in the dichotomous bid design (see reference 5 in the manuscript). 

The survey should really be in reverse. You first get the perception and knowledge of the farmers on FMD as you need them as independent variables. Then you need to make all of them understand the disease and its control by the vaccine whose attributes they need to understand before bidding because as per your background information, these farmers may not even know the vaccine. Then you pose the bidding questions. Apparently before bidding you did not explain these aspects to the farmers which makes one wonder whether they knew what they were bidding for.

Response: The first question in the questionnaire was used to verify whether the respondent did know the disease (appendix). It was only if the farmer correctly described the disease that he/she continued for the rest of the survey (this was clearly stated in line 194). Next the questionnaire describes the attributes of the vaccine the farmers are bidding for such as about its effectiveness, how it delivered and its price etc. So the concern that “the bidding questions are posed before the farmers know what they were bidding for” is not practiced. While the reverse order could have been also possible as commented, our intention was first to estimate the amount of the willingness to pay and then identify what factors could affect this willingness. 

L217: Change check to check; It is not clear how multicollinearity was checked

It is rare that a backward elimination of non-significant variables will result in a model with only significant variables.

Response: “checked’ corrected (L220)

Multicollinearity was checked using variance inflation factor (VIF) and VIF value above 10 was considered as indicator of presence of multicollinearity (see reference 22). This is made clear in the revised version (L219-222). 

The significant variables reported in our manuscript are those which remain significant after back ward elimination. We started with the maximum model (all variables), run the model, identify the variable with least significant p-value and drop this variable and run again, we continued until only significant variables were left in the model. 

L226: What does ‘inconsistent response’ mean?

 Response. If the answer for the second question cannot logically go with the answer for first question it is said inconsistent. For example. If the respondent answered he/she has no cattle for the first question, he/she should not answer the question about the breed of the cattle asked in the next question. 

L227: Who gave the ‘undetermined response’ for WTP?

Response: as choices were “yes”, “no” and “undermined” (see the annexed survey) few respondents answered “undetermined’ 

L230: The ‘majority’ here is only about half. How were the results of the other half? Explain from the table without repetition

Response: by majority we mean 50 +1. In this question the main interest whether livestock is main source of income or not and whether willingness to pay is strong when the main source of income is livestock. So the other sources of income (crop farming, trade, employment etc.) were categorized simply as others in the analysis. We indicated this in Table 2 of the current version. But we are afraid it will be irrelevant to list the proportion of farmers with each of these income sources 

Table 2: In the column, align contents conventionally; text aligned to the left and figures to the right. What is income from cattle and milk sale? Annual, monthly or what?

Response: the table formatted as suggested. The income from milk and cattle sales were asked for the preceding one year. This is indicated in the table as suggested. (Table 2) 

L255-260: It is not clear how the mean WTPs were calculated

Response: Here the willingness to pay was determined form a constant only (null) interval regression model. The constant of this model is the mean willingness to pay just as the constant term of a linear regression model is the mean value of the dependent variable except that the constant in interval regression is determined using maximum likelihood estimation instead of simple average of observed values of the dependent variable as in linear regression. This estimate of willingness to pay is without the use of the knowledge of socio-demographic and husbandry characteristics of the respondents. Later in Table 6 the willingness to pay was estimated taking into account socio-demographic and husbandry characteristics of the respondents using the best model built from the potential predictors (see the response for a similar comment below). The detail of the modelling can be found in reference 21 of the manuscript.

L262: The p should be >0.05 if no significant association

Response: The place and sign of the p value is changed for more clarity (L267) 

Table 4: Just indicate the reference category within the table by writing ref in the model coefficients column for the respective reference categories instead of having a footnote which is not clear

Response: modified as suggested (Table 4 and 5)

L261-265 and 272-278: Give the interpretation of the model coefficients, not just of the p value (significance). Considering the regression equation that arises from the results would help

Response: example of the interpretation of model coefficients has been given (L270-273). Interpreting all the coefficients would be monotonous as they can be easily seen in the table. 

L272-276: Remove unnecessary capitalization of words

Response; The unnecessary capitalizations are corrected (L281-282)

Table 6: It is not clear how the WTP estimates were derived

As explained above for a similar comment. Here willingness to pay estimates were derived form the best interval regression models built (table 4 for all respondents and table 5 for MCL and MO respondents separately). The best models were used to predict the willingness to pay for each respondent. From these individual estimates mean and median willingness to pay were calculated. The estimation using the mode has been described in L 213-216 and detail can be found in reference 21. 

L300: citation/reference? 

Response: Theory for demand is changed into law of demand and reference has been given (L309)

L311: What is ‘parametrized’?

Response: Fitting the model with data obtained from double bound dichotomous questioner survey 

L321-322: Correct the sentence

Response: the sentence is corrected (L330)

L328: What is ‘quality vaccine’ – you did not explain the vaccine to them

Response: the quality of vaccine was described to the farmers (in the appendix, question 2). Its efficacy was stated as 80%. According to OIE a standard potency FMD vaccine with protection percentage of 75% and above is considered as quality FMD vaccine. This has been also described L162-164

 L330: What is CLM?

Response: sorry it was to mean MCL; corrected (L328) 

L332: Change ‘tend’ to ‘tends’(L341)

Response: corrected 

L348: Avoid an abbreviation at the beginning of a sentence

Response: written in full as suggested (L358)

L362-365: Sentence is too long and needs rephrasing for clarity

Response: the sentence is split into two to make it clear (L373-375) 

L374: Correct the word knowledge

Response: corrected (L385) 

In the conclusion, WTP alone is not sufficient to make the conclusion that vaccine price can be covered by farmers fully without your knowledge on their ‘ability to pay’. You need to look at their profits or how they perceive profitability from their enterprises. As is, it is possible that WTP may represent valuation of vaccination rather than actual desire to pay the amounts

Response: The willingness to pay question was asked whether they are willing to pay for the vaccine at the stated prices. So If they answer yes, it is assumed that they think that it will be worthwhile (profitable) investment to pay that amount (see L 315-317 where farmers implied this rationale) and will pay if they are asked that amount for the prevention of the disease. 

L441: correct 20.20

Response: corrected (L452) 

Reviewer #2: Reviewer comments:

Excellent study design, data analysis, and overall manuscript. A few suggestions below. Suggest one more read through by a strong English speaker to remove typos. Identified some below, but there are probably more, and they distract from the quality of the paper.

Response: Serious proofreading has been done to minimize the grammar and typographical errors 

Line 35-36: should read “whose main livelihood is”

36 “livestock than those whose main livelihood is other than livestock,” – Rewrite, confusing

Response: the sentence is broken down for more clarity (Line 35-37). 

65 – typo – should read peste

Response: corrected (Line 66)

63 onwards – May be worthwhile to mention the role of diagnostic testing in FMD vaccination as an additional challenge – alluded to but not directly mentioned

Response: Yes, most of the time vaccination complicates diagnostic testing when the vaccines are not DIVA vaccines. But for FMD there are tests which can differentiate vaccinated and non vaccinated animals such as 3ABC ELISA which are based nonstructural proteins of the virus. So relatively this is not a major problem for FMD as compared to other important disease such as PPR where such tests or DIVA vaccines are yet not developed. 

78 – should read “vaccine is…” 

 Response: corrected (Line 80)

120 – contingent valuation is also appropriate in this situation because many people are not vaccinating.

Response : Yes, this is addressed in Line 124-3 on ‘poor adoption”

180 – should read “asked for”

Response: corrected as suggested (Line 184)

198 – should read farmers’

Response: corrected (Line 201)

199 – should read double-bounded (Line 201)

Response: corrected

347 – should read “is the main”

Response: corrected (Line356)

370 – should read “are willing to pay more”

Response: revised as suggested (Line 380)

373 – should read “pay more” 

Response: revised as suggested (Line 384)

381 – should read farmers’

Response. Corrected (Line 392) 

Response: corrected

385 – Should read farmers

Response: corrected (Line 396)

Gender – noticed the percentage of women respondents is very low. Do you think this is representative of the study population? Seems possible the selection of participants may bias against women if they are more likely to be at home. May be worthwhile to think more about this in the discussion.

Response: Clarification about this is added under sampling. The respondents were the head of households (Line 194) and in that setting the heads of the households are mainly males. 

---

## [Decision Letter · Decision Letter 1]

4 Sep 2020

PONE-D-20-17667R1

Farmers' willingness to pay for foot and mouth disease vaccine in different cattle production systems in Amhara region of Ethiopia.

PLOS ONE

Dear Dr. Jemberu,

Thank you for submitting your manuscript to PLOS ONE. After careful consideration, we feel that it has merit but does not fully meet PLOS ONE’s publication criteria as it currently stands. Therefore, we invite you to submit a revised version of the manuscript that addresses the points raised during the review process.

Many thanks for submitting your manuscript to PLOS One

Your manuscript was reviewed by the same two experts in the field as the original submission, and they have suggested some more minor revisions be made to it prior to acceptance.

If you could make these revisions, and write a brief response to reviewers, it will greatly expedite revision upon resubmission

I wish you the best of luck with your revisions

Hope you are keeping safe and well in these difficult times

Thanks

Simon

We look forward to receiving your revised manuscript.

Kind regards,

Simon Clegg, PhD

Academic Editor

PLOS ONE

Reviewers' comments:

Reviewer's Responses to Questions

**Comments to the Author**

1. If the authors have adequately addressed your comments raised in a previous round of review and you feel that this manuscript is now acceptable for publication, you may indicate that here to bypass the “Comments to the Author” section, enter your conflict of interest statement in the “Confidential to Editor” section, and submit your "Accept" recommendation.

Reviewer #1: All comments have been addressed

Reviewer #2: (No Response)

2. Is the manuscript technically sound, and do the data support the conclusions?

Reviewer #1: Yes

Reviewer #2: Partly

3. Has the statistical analysis been performed appropriately and rigorously? 

Reviewer #1: Yes

Reviewer #2: Yes

4. Have the authors made all data underlying the findings in their manuscript fully available?

Reviewer #1: Yes

Reviewer #2: No

5. Is the manuscript presented in an intelligible fashion and written in standard English?

Reviewer #1: Yes

Reviewer #2: No

6. Review Comments to the Author

Reviewer #1: (No Response)

Reviewer #2: Reviewer #2 Comments on PONE-D-20-17667_R1

September 3, 2020

23 vaccines

38 Remove “was also seen”

39-43 “perception that farmers would be reluctant to pay for FMD vaccine 41 is unprovable” – Be careful about how you word this. You have measured a distribution of WTP and some farmers may still have reluctance to pay. Review scientific method and do not talk about results suggesting something is unprovable.

“if official FMD vaccination is planned in the country, the vaccine cost can be covered” – Do not confuse WTP with what people will actually pay in a real life scenario. Higher WTP means respondents value the vaccine more but it does not necessarily mean “WTP scenario money” is exactly the same as real life money. Reword to say something like “market-oriented farmers with higher willingness to pay may be more likely to pay full cost if official FMD vaccination is planned in the country than mixed crop livestock farmers”

51 “the most important disease of livestock worldwide” – subjective statement. Reword.

56 Scandanavia

57 the case

63 onwards – I was not referring to DIVA vaccines but that without diagnostic testing, FMD vaccines can be poorly matched to the circulating serotype and therefore less effective. Is this applicable in Ethiopia? Address this as a constraint if so. https://www.ncbi.nlm.nih.gov/pmc/articles/PMC7067263/

66 replace cheap with affordable

77 pox, among others,

138 – knew – USE SPELL CHECK!! – It is reasonable to exclude farmers who are not familiar with FMD, but this will bias your WTP estimates up. You should then be using even more caution about the statements in lines 39-43. You have systematically removed farmers likely to be reluctant to pay then claimed farmers will cover costs. Be more conservative when wording conclusions.

296 Yes yes can also mean you truncated the distribution of the higher end of the WTP distribution curve.

302 – This suggests vaccines are valued but it does not necessarily mean farmers will pay the amount stated in a hypothetical scenario. Do not misinterpret as such.

364 – List threshold used when testing for multicollinearity

382 – See previous comments. Overstating your results again.

Supporting information – misspelled supplementary

Overall: Do not overstate results – be more careful in how the conclusions are worded. Please check English again and use spell check. Include dataset of WTP bids and relevant variables from the survey referenced in the text.

7. PLOS authors have the option to publish the peer review history of their article (what does this mean?). If published, this will include your full peer review and any attached files.

Reviewer #1: No

Reviewer #2: No

---

## [Author Response · Author response to Decision Letter 1]

10 Sep 2020

Response to the reviewer comments 

NB: the line numbers refer to the revised version of the Manuscript with track changes

23 vaccines

Response: corrected (L23)

38 Remove “was also seen”

Response: modified as suggested (L38)

39-43 “perception that farmers would be reluctant to pay for FMD vaccine 41 is unprovable” – Be careful about how you word this. You have measured a distribution of WTP and some farmers may still have reluctance to pay. Review scientific method and do not talk about results suggesting something is unprovable.

Response: the sentence modified to address the concern (L39-41). 

“if official FMD vaccination is planned in the country, the vaccine cost can be covered” – Do not confuse WTP with what people will actually pay in a real life scenario. Higher WTP means respondents value the vaccine more but it does not necessarily mean “WTP scenario money” is exactly the same as real life money. Reword to say something like “market-oriented farmers with higher willingness to pay may be more likely to pay full cost if official FMD vaccination is planned in the country than mixed crop livestock farmers”

Response: the interpretation of the results are revised to address the concern of overstating the results and the conclusions are restated accordingly (L39-44, 391-393). Moreover, in the discussion section, it has been discussed that contingent evaluation tends to overestimate the WTP as compared to actual market behavior and caution should be taken for practical application the results (L332-35). 

51 “the most important disease of livestock worldwide” – subjective statement. Reword.

Response: reworded as “ARGUABLY the most important disease of livestock worldwide” (L52)

56 Scandanavia 

Response: corrected (L57)

57 the case

Response: corrected (L58)

63 onwards – I was not referring to DIVA vaccines but that without diagnostic testing, FMD vaccines can be poorly matched to the circulating serotype and therefore less effective. Is this applicable in Ethiopia? Address this as a constraint if so. https://www.ncbi.nlm.nih.gov/pmc/articles/PMC7067263/

Response: Yes. There are multiple serotypes of FMD virus and vaccine matching is an issue in Ethiopia as well. This problem with FMD vaccine has been mentioned in this revision (L68-70). 

66 replace cheap with affordable 

Response: replaced as suggested (L67)

77 pox, among others,

Response: corrected as suggested (L79)

138 – knew – USE SPELL CHECK!! (Spelling checked as suggested (L 145))

 – It is reasonable to exclude farmers who are not familiar with FMD, but this will bias your WTP estimates up. You should then be using even more caution about the statements in lines 39-43. You have systematically removed farmers likely to be reluctant to pay then claimed farmers will cover costs. Be more conservative when wording conclusions.

 Response: as stated in the previous comment, the conclusions have been revised to address the concern with the over optimistic interpretation of the results (L39-44, 391-393). 

296 Yes yes can also mean you truncated the distribution of the higher end of the WTP distribution curve.

Response: correct. ‘yes’ ‘yes’ mean the upper bound for these respondents is higher than stated bids and shows the farmers are enthusiastic for the vaccine. 

302 – This suggests vaccines are valued but it does not necessarily mean farmers will pay the amount stated in a hypothetical scenario. Do not misinterpret as such.

Response: the statement here just compared the stated mean willingness to pay to the actual price and does not necessary imply that they will pay that amount. 

364 – List threshold used when testing for multicollinearity

Response: already mentioned in the material and method part i.e. VIF equal or greater than 10 was considered as indicator of multicollinearity (L225-26) 

382 – See previous comments. Overstating your results again.

Response: the conclusion revised to avoid the overoptimistic interpretation (L391-393)

Supporting information – misspelled supplementary

Response: modified as supporting information (S1)

Overall: Do not overstate results – be more careful in how the conclusions are worded. Please check English again and use spell check. Include dataset of WTP bids and relevant variables from the survey referenced in the text.

Response: the overoptimistic interpretation of results are revised and the conclusion are restated accordingly (L39-44, 391-393). Spell check has been done to avoid language errors and the data set for the WTP are included as supporting information (see S2)

---

## [Editor Report · Decision Letter 2]

15 Sep 2020

Farmers' willingness to pay for foot and mouth disease vaccine in different cattle production systems in Amhara region of Ethiopia.

PONE-D-20-17667R2

Dear Dr. Jemberu,

We’re pleased to inform you that your manuscript has been judged scientifically suitable for publication and will be formally accepted for publication once it meets all outstanding technical requirements.

Kind regards,

Simon Clegg, PhD

Academic Editor

PLOS ONE

Additional Editor Comments:

Many thanks for resubmitting your manuscript to PLOS One

As you have addressed all the comments, and the manuscript reads well, I have recommended it for publication

You should hear from the Editorial Office soon

It was a pleasure working with you and I wish you all the best for your future research

Hope you are keeping safe and well in these difficult times

Thanks

Simon

---

## [Editor Report · Acceptance letter]

23 Sep 2020

PONE-D-20-17667R2 

Farmers’ willingness to pay for foot and mouth disease vaccine in different cattle production systems in Amhara region of Ethiopia 

Dear Dr. Jemberu:

I'm pleased to inform you that your manuscript has been deemed suitable for publication in PLOS ONE. Congratulations! Your manuscript is now with our production department. 

Kind regards, 

on behalf of

Dr. Simon Clegg 

Academic Editor

PLOS ONE